# Microplastic in Food and Water: Current Knowledge and Awareness of Consumers

**DOI:** 10.3390/nu14224857

**Published:** 2022-11-17

**Authors:** Klaudia Oleksiuk, Karolina Krupa-Kotara, Agata Wypych-Ślusarska, Joanna Głogowska-Ligus, Anna Spychała, Jerzy Słowiński

**Affiliations:** 1Department of Epidemiology, Faculty of Health Sciences in Bytom, Medical University of Silesia in Katowice, 40-055 Katowice, Poland; 2Department of Environmental Health, Faculty of Health Sciences in Bytom, Medical University of Silesia in Katowice, 40-055 Katowice, Poland

**Keywords:** microplastic, food, water, knowledge, consumers

## Abstract

In recent years, the widespread of microplastics in the food chain and environment became a topic of much research. This article focused on the knowledge and awareness of people with higher education levels—mostly young ones. The aim of this study is to analyze to what extent consumers know about and are aware of the source of microplastics, the level of exposure, and potential health hazards connected to the contamination of food and water with microplastics. The test group, consisting of 410 people, is mostly able to correctly characterize what microplastics mean and knows its sources. A majority of the group is aware of potential presence of microplastics in water; however, the knowledge about contamination of other elements of the environment seems to be gradually lowering. The majority of the people taking part in the research know that microplastic might be present in foods, and they are aware that after entering the human body, it might accumulate in internal organs. Moreover, when asked about potential health hazards, the group chose mostly tumors and gastrointestinal disorders, while disorders of the reproductive system were chosen less frequently. Consumers’ knowledge regarding the sources and health hazards of microplastics seems to be more common among women, in groups living in cities and among people who studied physics-related subjects and medicine.

## 1. Introduction

Plastics can be used in a wide variety of ways, giving the society multiple benefits from its usage. However, recently it is more and more debated upon whether it is wise to withdraw it from common usage and limit its release to the water bodies. The main point of interest here is microplastics, which are tiny (less than 5 mm in diameter) pellets made of plastic, or small pieces being the result of plastic’s breakdown. Their content and structure are the cause for new environmental hazards [1,2]. As a result, we can highlight primary microplastics, used as a resource in the plastic industry, in pharmaceutics and cosmetics, as well as abrasive blasting. Secondary microplastics come from the breakdown of bigger plastic particles, during the degradation process [3,4,5].

Considered anthropogenic debris, microplastics are found in the entire world, especially in water bodies. Although it is a new field of research, their negative impact on environment and potential health hazards for both animals and humanity became the scope of research for many experts.

The presence and buildup of microplastics is a danger to environmental balance, watery environments, lasting and safety of food, and eventually, human health [3,6,7]. Plastics can both absorb and adsorb toxic inorganic substances and organic compounds, which can penetrate through the tissue of sea creatures. So far, it was believed that the soil and settlements are the biggest sorbents in the environment; however, plastics accumulate contaminants even faster. Moreover, the overgrowth on pieces of plastic increases the surface capable of absorbing contaminants, through sorption or accumulating in plant tissue. Those substances are called hydrophobic organic contaminants, and include polycyclic aromatic hydrocarbons, polychlorinated biphenyls, and DDT. Those substances are known for remaining unchanged for extended periods of time, are widely spread in the environment, accumulate in living tissue, and most importantly of all, are toxic to animals and humans. Microplastics can adsorb and concentrate HOCs and organochlorine pesticides on a much higher level on their hydrophobic surface. Other than HOCs, on the surface of microplastics one might also find heavy metals, such as cadmium, zinc, lead, nickel, and chemical compounds used to manufacture plastics [8]. Plastics floating on the surface undergo degradation caused by UV rays, as well as mechanical and microbiological ones. The cold environment of the sea environment does not work in photooxidation’s favour, so plastics in water bodies are going to remain for longer in the environment. Huge availability of oxygen, high temperatures, and presence of the sun allow the fast degradation of plastics on the beaches. Over time, their fragments become fragile, and the material becomes yellow and fragmentated due to waves and abrasion. Its fragments might accumulate in gastrointestinal tracts of sea creatures as a result of swallowing. Microplastics can penetrate into the tissue and accumulate there [9].

It is not yet determined what the risks for humans connected to the dietary exposure to microplastics are. Humankind is exposed to microplastic through contact, ingestion, and inhalation. The research shows that the most common exposure comes from ingestion, so the main source of exposure is diet [6,10,11]. Other sources point out to dust, soil, textiles, industry, and general waste [12,13]. Being on the top of the food chain, humanity consumes both plants and meat, including seafood, which earlier absorbed microplastic into their systems by various means. It is also possible that humanity absorbs microplastics through sea salt and food that went through the production process and contains pieces of packaging and machinery. It was confirmed with stool samples of multiple countries’ residents. Microplastic is also present in drinks sold in plastic bottles.

Microplastics may accumulate and exert localized particle toxicity by induction or enhancement of an immune response both through inhalation and ingestion. Chemical toxicity might be occurring as a result of the localized leaching of component monomers, endogenous additives, and adsorption of environmental pollutants. Because of the accumulative effect that could occur, chronic exposure is anticipated to be of greater concern. This is expected to be dose-dependent [14]. Microplastics may occur in seafood, processed food, and beverages (sugar, beer, and salt) [15,16]. Given the earlier mentioned presence of microplastics in the marine environment, an expected route of human exposure is through seafood. Seafood provides approximately 20% of their animal protein intake to almost 3 billion people worldwide. That makes it also one of the most important foods consumed globally; however, it can also be a source of contaminants, such as PCBs and dioxins [14]. Globally, fish provides approximately 4.3 billion people with 15% of their animal protein intake. In the case of fish, the occurrence of microplastics in the gastrointestinal tract is not an evidence for human exposure, as this organ is usually not consumed. There is potential for the leaching and accumulation of associated chemical contaminants in edible tissue, post-microplastic ingestion. Consumption of the skin or gill tissue could present a direct route of human exposure to microplastics [14]. Other than seafood, microplastics were reported in other foods. The presence of synthetic microfibers (minimum 40 μm in length) and fragments (mostly 10−20 μm in size) was reported in honey and sugar, beer, and sea salt. The water source may also be a source of contamination [14].

The common occurrence of microplastics in food chains and the environment caused it to be the center of many researches’ studies [6,17]. It is widely analyzed; however, there is still not much research concerning the consumers’ knowledge and awareness of the microplastics’ presence in food and water, and its consequences.

Taking this into account, in this study, we decided to conduct an awareness survey of a population of students from the Silesian region (Poland) regarding the potential impact of microplastics on the risk of negative health effects. The choice of the study population was not random and was based on several important considerations. It is worth mentioning that the Silesian population living in a highly urbanized and industrialized area is particularly vulnerable to environmental health risks. The fundamental reason was that young consumers (and such are the vast majority of students) are expected to be more consumer- and health-conscious. The awareness in question at the same time is a determinant of their own health status, but in the near or long term, it will also shape the health status of their offspring [18,19]. Sufficient consumer awareness makes it possible to understand the potential negative consequences of microplastic contamination of food, and to make informed, health-related consumer decisions. Young people, both educated in the field of medical and health sciences, but also educated in other areas of knowledge, should have a special perception of the health risks resulting from exposure to various xenobiotics, as evidenced by the large number of studies devoted to this issue. Studies are available on the perception of health risks resulting from exposure to air pollution or exposure resulting from heavy metals in food and water [20,21,22]. Since environmental pollution by microplastic particles is a relatively new issue, yet currently one of the most intensively researched, it was felt that investigating young people’s knowledge of oral exposure to microplastics was warranted and would add to the, still scarce, body of knowledge in this area [23,24,25]. Finally, it is worth mentioning that the college-educated population is seen as particularly attractive for yet another reason. Due to the body of knowledge they present, the specificity of their duties resulting from receiving education, and familiarity with a research tool such as a questionnaire, students show a willingness to cooperate, see the validity of conducting scientific research, and the possibility of using their results to improve quality of life in many dimensions. Thus, they are a particularly cooperative study group, which was seen as an additional potential in the selection of the study population.

The aim of this study was to research the current knowledge and awareness of aforementioned consumers regarding sources, exposure, and health hazards connected to microplastics’ presence in water and foods, especially its impact on internal organs, metabolic processes and reproductive functions. After having completed the necessary reading and having in mind the goal of the paper, it was hypothesized:Consumers do know what microplastic is and what its source is, and are aware that it is a danger mostly to water bodies.Consumers are aware that microplastic might be a risk when it comes to food safety.Consumers are not aware of potentially negative impact coming from exposure to the microplastics through the gastrointestinal tract.Consumers’ knowledge regarding the sources of the exposure and health hazards looks better among the group of people educated in the field of physics and medicine.Consumers’ knowledge regarding the sources of the exposure and health hazards is more common among women and people living in cities.

End results enable the judgement of the consumers’ awareness about the hazards coming from microplastics’ presence in food and water and indicated the need for educating young people about the researched field. Formation of appropriate behavioral change may lead to a shift in people’s behavior in terms of plastic consumption if they become aware of the environmental risk of plastics.

## 2. Materials and Methods

### 2.1. Study Organization and Eligibility Criteria

In order to conduct the research, a questionnaire was used. A computer-aided web interview (CAWI) was chosen, and the questionnaire was shared via Google Forms. The distribution of the survey was conducted using its link, which was then sent via WhatsApp, email, and Facebook with specificity of only Poland. It was planned for the time from August to October 2021; however, due to the low amount of answers, it was decided that it shall be extended to June 2022.

The respondents took part in the questionnaire voluntarily and anonymously and were informed about the point of the questionnaire and usage of their answers solely for scientific purposes. The selection of respondents was non-random, as the snowball method was used.

In order to take part in the questionnaire, it was required to be of age above 18, live in Silesia (Poland) (Figure 1), and be a student. The questionnaire was filled correctly 410 times. The necessary sample size was calculated, depending on the university, using a finite population formula and was 384 students. It was estimated that a sample of 410 students would be sufficient and representative of the Silesian region in Poland. It was assumed, according to the Central Statistical Office (CSO) report, that the ratio of people studying in Silesia is 106,411. Accordingly, on the formula: Nmin = NP ∙ (α^2^ ∙ f(1 − f)) ÷ NP ∙ e^2^ + α^2^ ∙ f(1 − f), where: Nmin—minimum sample size; NP—size of the population from which the sample is drawn; α—confidence level for the results; f—size of the fraction; and e—assumed maximum error. The minimum sample size of students was calculated for the population of Silesia (Poland), which was 384 students (α = 0.95; f = 0.9; and e = 0.05). Based on these calculations, the collected group of students was considered representative. Relationships between age, gender, and basic metric data between groups were examined, and the statistical test showed no relationship *p* > 0.05.

This research project complied with the university’s ethical guidelines. All data were coded with appropriate symbols preventing the identification of individuals participating in the study in accordance with the Act of 29 August 1997 on the Protection of Personal Data (Journal of Laws 1997 No. 133 item 883). The study design, in the light of the Act of December 5, 1996 on the professions of physician and dentist (Journal of Laws of 2011 No. 277.item 1634 as amended), is not a medical experiment and does not require the evaluation of the Bioethical Committee of the Medical University of Silesia in Katowice, since the study con-centered on the experiences of the participants.

### 2.2. Study Procedure and Research Tool

The questionnaire consisted of demographics and 26 questions. Based on the demographics answers, the respondents were classified based on their university major: humanities; engineering and technical; medical and health; social sciences; and natural sciences. Repeatability of the responses was examined by distributing the questionnaire twice to a random sample of 20 people. The second round of the survey took place two weeks after the first questionnaires were collected. The same questionnaire form was distributed again by electronic agreement to those who previously responded to the survey invitation. 

The research tool of the study was the author’s anonymous questionnaire, which was validated for reliability, correctness, and relevance. Reproducibility was assessed based on the magnitude of the percentage of concordant responses to the same questions in both rounds and by calculating the kappa statistic.

In order to assess the reproducibility of the results obtained with the questionnaire used, the value of the χ parameter (Cohen’s kappa) was calculated for each questionnaire question (results obtained in the baseline and retest). A total of 78.3% of the questions obtained a very good agreement Kappa > 0.75.

People taking part in the questionnaire were also divided based on their gender, age (19–24 years, 25–30 years, 31–40 years, and >40 years), activity in the labor market, and place of living. The proper questionnaire asked questions about microplastics, their source presence in parts of the environment, water and food, their metabolism in the living organism, as well as possible health hazards coming from their presence in the human body. The last question asked about possible ways of reducing the amount of microplastic exposure. The questionnaire consisted of closed questions (of singular and multiple choice).

### 2.3. Statistical Compilation

The collected material was subjected to statistical analysis, and in order to examine the relationship between the variables, non-parametric Chi2 and V-square tests of independence were used. The strength of the relationship (correlation) was checked using the V-Cramer coefficient. A significance level of α = 0.05 was used for the analyses, and the statistically significant results were defined by the *p*-value < 0.05. Statistical analysis of the collected material was performed using the Statistica program (Software TIBCO Software Inc.—2017, version 13, Palo Alto, CA, USA).

## 3. Results

Table 1 presents characteristics of the study group. The majority of the group consisted of people between the age of 19 and 25 (62.4%). Almost 65% of the group consisted of women, and over 35% of men. Among women, over 65% were of age between 19 and 24, and around 20% of age from 25 to 30. Among men, 60% were of an age between 19 and 24, and 32.6% between 25 and 30.

Around 41.2% of the people taking part in the questionnaire were studying medicine-related majors, and around 35.6% picked social sciences. Among women the majority study medicine-related sciences (52.3%) and social sciences (33.1%), while only 2.6% follow the natural science courses. Among men, 40.3% studied social sciences, 27.8% engineering/technical studies, and only 3.5% were in natural sciences.

Over half of the group is active in the labor market (55.4%). Over 42% of the group never worked before, and 2.4% receive a pension, have paid internships, or work in a part-time job. The proportions were similar both among men and women. 

The majority of respondents live in cities with over 100 thousand inhabitants (54.6%). Over 36% of participants live in cities with a number of inhabitants below 100 thousand (36.4%). Only 9% of the group lives in villages and smaller towns.

The people taking part in the research were asked a question: “What are microplastics?” Over 68% of participants answered correctly, that microplastics are tiny pellets made of plastics and small pieces formed during the breakdown of plastics, and that they might be a danger to the natural environment. Over 17% thought that microplastics were synthetical polymers used in cosmetic products. A total of 8.3% chose the answer that microplastics are microscopical nylon threads that do not threaten the environment. A little bit over 6% chose that microplastics are soft polymers of various shapes that dissolve in water (Figure 2).

Figure 3 shows the percentage of answers regarding the source of so-called primal microplastic. Only less than half of the group (49.5%) answered correctly that the main source is synthetic textiles and car wheels during the drive. One in five of the respondents believed that the main source is city dust. A total of 18% decided that the main source is in cosmetics, while 12.2% that it is paint chipping from buildings and road signs.

The research group was asked to pick the elements of the environment where it is likely to find microplastics. Over 90% knew that microplastics may occur in water. Around 65.6% of the group knew that it can be found in soil, while 60% were aware of its presence in the air. The least picked answer was living organisms (Figure 4).

Table 2 shows the state of answers regarding the presence of microplastics in water, food, and the accumulation of microplastics in internal organs. The questions in Table 2 called for only one answer. The field of study of the group was taken into consideration.

Almost 54% of the study group thought that microplastic might appear in tap water (53.9%). A total of 28.5% thought the opposite, and 17.6% could not answer the question. Around 83% of natural sciences’ students were aware of the occurrence; however, the group was relatively small—consisted of only 12 students, and 75.1% of people studying medicine-related sciences answered the same. The lowest percentage of correct answers was among students of social studies, where only 3.1% knew the answer.

In the question about potential presence of microplastics in bottled water, 42.7% answered that the contamination is possible, and 38.5% believed the opposite. In this matter the biggest percentage of correct answers appeared in groups studying natural sciences (75%) and medicine (65.1%).

Almost 87% (86.6%) of the study group decided that microplastics might be present in food. Only 4% did not think so, and 9.7% could not answer the question. Around 92% of people majoring in engineering, 89% of people studying medicine, 86% of social studies’ students, over 83% of natural sciences’ students, and 69% of humanities students answered correctly.

In the question regarding the vegetables most likely to be contaminated with microplastics, the students picked mainly root (37.1%) and brassica (28.8%) vegetables, which sums up to over 65% of all the answers. The remaining students picked bulb vegetables (17.3%) and leguminous vegetables (8.5%) as the ones contaminated the most, and 8.3% decided that vegetables are not contaminated at all. The correct answer, that the root vegetables are the ones most likely to be contaminated, was picked by 47% of students majoring in medicine-related fields.

Almost 80% of the study group knew that microplastics can accumulate in internal organs in the human body. The knowledge of this matter looked the same among the sub-groups, with an exception of students of humanities, where only 65.5% knew the answer.

Table 3 presents the study group’s knowledge regarding the presence of microplastics in various foods and potential health hazards caused by them. The questions of Table 3 were multiple choice, and again, the field of study was taken into consideration. The respondents, no matter their major, picked mostly marine fish, seafood, and freshwater fish as the ones most likely to be contaminated. Among potential health hazards they picked cancers, gastrointestinal diseases, inflammation, and thyroid diseases. Disorders of the reproductive system were chosen the least. It was picked by 58.3% of natural sciences’ students, 46.7% of medicine and health students, 31.5% of engineering students, 29.5% of social studies’ students, and 17.5% of humanities students.

Table 4 presents results of independence tests between knowledge of the topic of microplastics and data characterizing the study group.

The analysis of the results of independence tests showed that there is a correlation between:-the knowledge of what microplastics are and participants’ gender; women were aware of it more often than men,-the knowledge of what microplastics are and university major of participants; over 91% of science students, 83% of medicine and health students, and 60% of social studies and humanities students knew the correct answer, while among engineering students it was merely 40%,-the knowledge of what microplastics are and participants’ place of living; the best results occurred among students living in cities with more than 100 thousand inhabitants,-the knowledge regarding the presence of microplastics in tap water and gender of participants; women were more likely to know that than men,-the knowledge regarding the presence of microplastics in tap water and university major; over 83% of physics and natural sciences students and over 75% of medicine and health students knew that microplastics may occur in tap water. Only 30% of students of remaining majors knew the correct answer,-the knowledge regarding the presence of microplastics in tap water and place of living; over 60% of group living in the cities of over 100 thousands in habitants, 30% of group living in the cities with population below that number and 10% of group living in villages knew the correct answer,-the knowledge regarding the presence of microplastics in tap water and professional status; around 60% of working people knew the correct answer, while amongst unemployed it was only 40%,-the knowledge regarding the presence of microplastics in bottled water and participants’ gender; women are more aware of its presence in bottled water than men,-the knowledge regarding the presence of microplastics in bottled water and university major; over 75% of physics and natural studies’ students and over 65% of medicine and health studies’ students knew the correct answer, while among other majors it was only 20%,-the knowledge regarding the presence of microplastics in bottled water and place of living; 59% of group living in cities of population over 100 thousands, 31% of group living in cities below 100 thousands, and 10% of people living in villages knew the right answer,-the knowledge regarding the contamination of vegetables and gender of participants; women mostly picked root and brassica vegetables (44% and 26%, respectively), while among men the most common answer was brassica (33%), bulb (26%), and leguminous (24%) vegetables,-the knowledge regarding the contamination of vegetables and university major; students of physics and nature sciences decided that the most likely type to become contaminated is brassica (over 66%), students of medicine and health studies picked root (over 47%) and brassica (31%) vegetables, engineering students chose brassica and bulb vegetables (31%), and students of humanities and social studies picked root vegetables (over 30%).

Coefficient of correlation indicates low (0.1 ≤ R < 0.3) and average (0.3 ≤ R < 0.5) levels of correlation.

## 4. Discussion

The presence of microplastics in the environment and food chain is becoming researched more frequently. Although the impact microplastics have on the world is becoming recognized by scientists and society, it is risky to evaluate the influence of dietary microplastics so far. The most important problem needing further research is the case of food safety and the possible negative impact of microplastics on human health [6,8,26]. The recent findings of those occurring in food and water and the evidence of human exposure to plastics, as well as the long-term health effects of exposure being largely unknown, makes the entire topic more disturbing [25]. The awareness of the risk to food safety and the meaning of keeping food and water hygiene is important to all groups in society; however, the most exquisite group consists of young consumers, since it is expected of them to be more knowledgeable regarding health and consumption. Ojinnaka and Aw described in their research, made to assess the consumers’ perception of the control system and environmental and food safety threats of micro-nanoplastics, a few hypotheses. One of them says that there is a low consumer perception of the environmental and food safety threats of micro-nanoplastics. The hypothesiss was partially proven, since there was some general awareness of the environmental threat; however no clear conclusion on the food safety threat was reached [8]. In a questionnaire taken among the polish students, it was shown that over 68% of respondents know what microplastics are. However, the knowledge about sources of microplastics gave a bit less impressive results. Synthetic textiles and usage of car tires were picked as the main source only among 49.5% of the research group. In order to access the knowledge of the exposure of society to microplastics, the group was asked in what parts of the environment one can find microplastics. The majority of the group was aware of microplastics’ presence in waters; however, the knowledge regarding the other parts of the environment, i.e., soil, air, and living organisms, was considerably lower. Ojinnaka and Aw also proved that there is a possibility that the awareness level is related to the level of education, as most respondents have higher education, but it was not possible to statistically prove it [8]. In the research conducted in this article, it was proven that there is a relation between the knowledge of what microplastics are and the university major of respondents. The correct definition of the word microplastics was chosen by 91% of physics and natural sciences students, over 83% of medicine and health students, and around 60% of social studies and humanities students. Only 40% of engineering and technical major students know correctly what microplastics are. It was also concluded that there is a relation between the gender of respondents an their knowledge of the topic. Women were more likely to give the correct answer than men. The university major turned out to be a factor influencing the awareness of potential food safety breaches. People studying physics, natural studies, medicine, and health studies were more aware of the presence of microplastics in both tap and bottled water than students of the remaining fields. People studying medicine and health-related studies seemed to also know more about the vegetable contamination by microplastics. Around 47% of them answered that the types of vegetable mostly prone to such contaminations are root vegetables.

Despite the diversity of food in different regions of the world, only a small amount of studies were undertaken on the presence of microplastics in food. The biggest amount of studies focuses on analyzing the concentration, materials, morphology, and size of microplastics in salt, fish, and shellfish [19]. With microplastics total diet studies entirely absent in the academia, some exposure estimations identify drinking water and seafood as the main microplastics dietary sources. It is found also in foods such as honey, milk, or beer. According to Ojinnaka and Aw, 82% of respondents picked oceanic produce and cosmetics as the main sources of microplastics and nanoplastics [8]. Similar results occurred in this research. Respondents mostly picked marine life and seafood as the main type of food contaminated by microplastics. Although a large number of studies on seafood such as fish and shellfish exist, estimating the exposure to microplastics via food consumption is difficult. The reason could be assigned mostly to the lack of studies on other food items. In order to properly recover microplastics from various food sources, there are still improvements to be made in analytical methods undertaken. That makes a quantitative comparison of different studies challenging [27]. Several institutions and researchers believe that the current knowledge does not supply enough data to complete a risk characterization of microplastics diet-wise [22].

Usman et al. say that, regarding the toxicodynamic of these food pollutants, it is expected that their action mechanism in humans is most likely to be similar to that observed in animals [21]. Therefore, it is to be expected that the microplastics could affect many molecular pathways [23] and disrupt the genetic expression of oxidative stress control. The consequence of such mechanisms can include alterations and changes in the oxidative stress, immune response, genomic instability, endocrine system alteration, neurotoxicity, reproductive abnormalities, embryotoxicity, and transgenerational toxicity [24]. In the questionnaire taken for the sake of this paper, the most common consequence of exposure to microplastics picked by the respondents was cancers, gastrointestinal diseases, inflammation, and thyroid diseases. The diseases of the reproductive system was way less popular as an answer to the question. The main group aware of the connection between the exposure and disruption of the reproductive functions was students of nature sciences, medicine, and health science. It was picked the least among humanities’ students.

It is obvious that the increase in plastic consumption is a cause for concern for both researchers and policy makers, no matter whether it has a negative impact on the environment or not [28]. It is, however, still not clear if people in different socioeconomic groups care about plastic pollution, neither if they are willing to take action against plastic consumption amongst their communities. This raises different questions for further researchers. Are people aware of the plastic problem? What do people think about it? How do people define pollution? Do they care about the impact of plastic on the environment? Do they take any action in order to address this problem? Does economic or food safety prevail when making consumer choices? It is recommended to conduct further research, possibly covering wider groups of the population, in order to answer these questions.

## 5. Strengths and Limitations

The study focused on a group of young, studying people, and thanks to the calculation of minimum sample size, the results remain representative of the Silesian region (Poland). In addition, the results did not show differences by gender and age, and basic metrics thus did not affect the variation between groups and allowed to expose the variation in respondents’ knowledge depending on the field of study. Thus, this is an important piece of information indicating the need to also educate students about the dangers of microplastics in the environment. Admittedly, the survey protocol does not make it possible to determine whether such education was carried out, and if so, to what extent. However, it is worth noting this aspect in future research as well. The survey was conducted using the internet, which makes it difficult to control how the questionnaires were filled out. It is also unclear whether there is a selection phenomenon in the survey, resulting from the entry of people with a special interest in the issues discussed. On the other hand, even if this were the case, it indicates incomplete knowledge of those who may possibly be more interested in the issue of microplastic pollution. Thus, it is worth emphasizing once again the added, utilitarian value of the present study, indicating the necessity of including in the scope of research the topic of microplastics in various dimensions appropriate to the chosen field of study. At the time of editing the present manuscript, a search of online databases indicated the availability of a small number of papers on the study of knowledge about microplastics and their effects on health in different segments of the population. However, there is no doubt that awareness, especially among young adults, is of particular interest to researchers around the world. Research on knowledge of microplastics is growing, and there are more and more of it. As the analysis for the keyword “knowledge of micro-plastics” in PubMed shows, only in 2018 was there a marked increase in interest in this topic, but still, the number of studies on knowledge does not exceed 300 per year. Hence, in the near future, the number of published research papers on this topic may increase significantly.

## 6. Conclusions

Consumers do know what microplastic is and what its source is, and are aware that it is a danger mostly to water bodies.A huge number of respondents know that microplastic may occur in food and pick mostly fish and seafood as the source.Respondents also know that microplastic might accumulate in internal organs. When it comes to potential health hazards connected to exposure to microplastics, they mostly pick gastrointestinal diseases, inflammation, and thyroid diseases. So the hypothesis that consumers are not aware of the potential negative impact coming from exposure to the microplastics through the gastrointestinal tract was not confirmed. The reproductive system disease is chosen way less frequently.Consumers’ knowledge regarding sources of gastrointestinal exposure and its negative consequences seems to be more common among people educated in physics and medicine.Consumers’ knowledge regarding the sources of the exposure and health hazards is more common among women and people living in cities.

## Figures and Tables

**Figure 1 nutrients-14-04857-f001:**
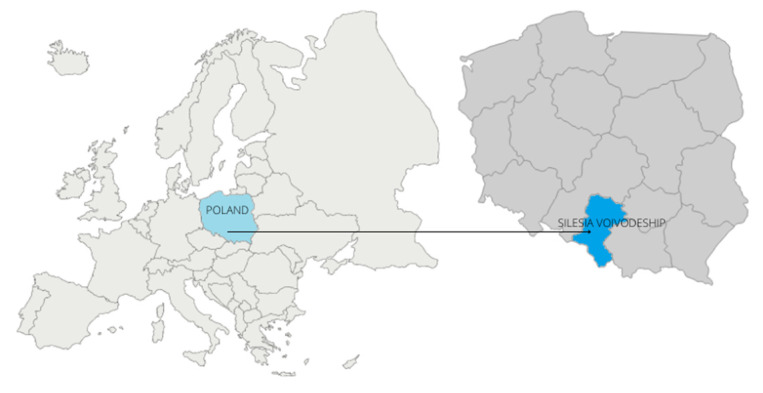
The place of the survey.

**Figure 2 nutrients-14-04857-f002:**
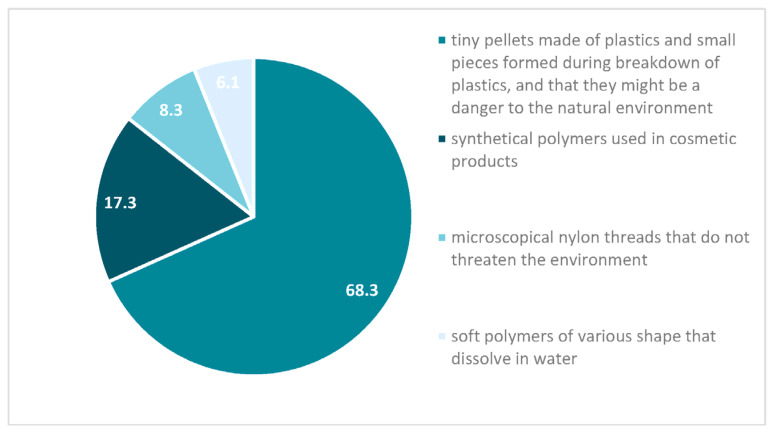
Respondents’ answers to “What are microplastics?” (%).

**Figure 3 nutrients-14-04857-f003:**
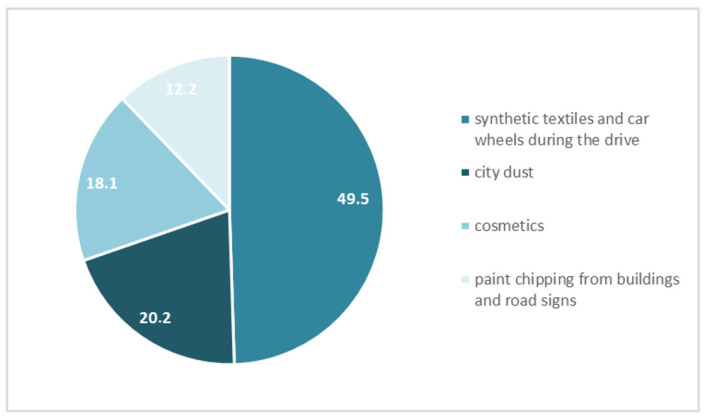
Respondents’ answers to “What is the source of so-called primary microplastics?” (%).

**Figure 4 nutrients-14-04857-f004:**
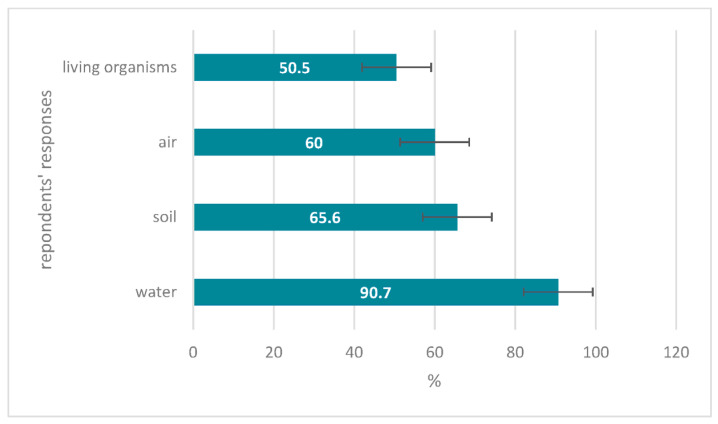
Respondents’ answer to “In what elements of the environment it is possible to find microplastics?” (%).

**Table 1 nutrients-14-04857-t001:** Characteristics of the study group.

Variables	Women % (*n*)	Men % (*n*)	Total % (*n*)
Age group:			
19–24 years	65.4% (N = 174)	60.0% (N = 82)	62.4% (N = 256)
25–30 years	19.9% (N = 53)	32.6% (N = 47)	24.4% (N = 100)
31–40 years	10.9% (N = 29)	7.6% (N = 11)	9.8% (N = 40)
>40 years	3.8% (N = 10)	2.8% (N = 4)	3.4% (N = 14)
Science fields:			
humanities	6.8% (N = 18)	7.6% (N = 11)	7.1% (N = 29)
engineering and technical	5.3% (N = 14)	27.8% (N = 40)	13.2% (N = 54)
medical and health	52.3% (N = 139)	20.8% (N = 30)	41.2% (N = 169)
social sciences	33.1% (N = 88)	40.3% (N = 58)	35.6% (N = 146)
natural sciences	2.6% (N = 7)	3.5% (N = 5)	2.9% (N = 12)
Job status:			
working	55.6% (N = 148)	54.9% (N = 79)	55.4% (N = 227)
not working	41.7% (N = 111)	43.0% (N = 62)	42.2% (N = 173)
other	2.7% (N = 7)	2.1% (N = 3)	2.4% (N = 10)
Accommodation:			
City ≤ 100 k inhabitants	29.3% (N = 78)	49.3% (N = 71)	36.4% (N = 149)
City > 100 k inhabitants	59.4% (N = 158)	45.8% (N = 66)	54.6% (N = 224)
small town	11.3% (N = 30)	4.9% (N = 7)	9.0% (N = 37)
Total	64.9% (N = 266)	35.1% (N = 144)	100% (N = 410)

**Table 2 nutrients-14-04857-t002:** The state of respondents’ knowledge about the occurrence of microplastics in water, food, and health effects.

Variables		Science Fields % (N)					
		Humanities	Engineering and Technical	Medical and Health	Social Sciences	Natural Sciences	Total
Is it possible to find microplastics in tap water?	yes	9 (3.1)	20 (37.0)	12 (5.1)	55 (37.7)	10 (83.3)	221 (53.9)
	no	15 (51.7)	23 (42.6)	17 (10.1)	62 (42.5)	-	117 (28.5)
	I don not know	5 (17.5)	11 (20.4)	25 (14.8)	29 (19.9)	2 (16.7)	72 (17.6)
Is it possible to find microplastics in bottled water?	yes	6 (20.7)	15 (27.8)	110 (65.1)	35 (24.0)	9 (75.0)	175 (42.7)
	no	19 (65.5)	29 (53.7)	35 (20.7)	74 (50.7)	1 (8.3)	158 (38.5)
	I don not know	4 (13.8)	10 (18.5)	24 (14.2)	37 (25.3)	2 (16.7)	77 (18.8)
Is it possible to find microplastics in food?	yes	20 (69.0)	50 (92.6)	150 (88.8)	125 (85.6)	10 (83.4)	355 (86.6)
	no	2 (6.9)	1 (1.0)	4 (2.4)	7 (4.8)	1 (8.3)	15 (3.7)
	I do not know	7 (24.1)	3 (5.6)	15 (8.9)	14 (9.6)	8.3% (N = 1)	40 (9.7)
Which type of vegetable is the most likely to be contaminated by microplastics?	brassica	5 (17.2)	17 (31.5)	52 (30.8)	36 (24.7)	8 (66.6)	118 (28.8)
	bulb	5 (17.2)	17 (1.5)	18 (10.7)	29 (19.9)	2 (16.7)	71 (17.3)
	root	9 (31.0)	12 (22.2)	80 (47.3)	49 (33.6)	2 (16.7)	152 (37.1)
	legumes	3 (10.3)	8 (14.8)	5 (3)	19 (13.0)	-	35 (8.5)
	they are not contaminated	7 (24.1)	-	14 (8.3)	13 (8.9)	-	34 (8.3)
Is it possible that after entering the human body, microplastics can accumulate in internal organs?	yes	19 (65.5)	44 (81.5)	138 (81.7))	116 (79.5)	10 (83.4)	327 (79.8)
	no	2 (6.9)	4 (7.4)	2.3% (N = 4)	11 (7.5)	-	21 (5.1)
	I do not know	8 (27.6)	6 (11.1)	16% (N = 27)	19 (13.0)	2 (16.6)	62 (15.1)
Total		29 (100)	54 (100)	169 (100)	146 (100)	12 (100)	410 (100)

**Table 3 nutrients-14-04857-t003:** The state of respondents’ knowledge about the occurrence of microplastics in food products and the health effects in the human body caused by microplastics.

Variables		Science Fields *n* (%) *				
		Humanities	Engineering and Technical	Medical and Health	Social Sciences	Natural Sciences
The occurrence of microplastics in food products	marine fishes	20 (69.0)	52 (96.3)	163 (96.4)	135 (92.5)	12 (100.0)
	freshwater fish	11 (38.0)	17 (31.5)	132 (78.1)	80 (54.8)	9 (75.0)
	seafood	11 (38.0)	30 (55.6)	146 (86.4)	83 (56.8)	10 (83.3)
	sea salt	5 (17.2)	19 (35.2)	89 (52.7)	46 (31.5)	8 (66.7)
	honey	1 (3.4)	9 (16.7)	52 (30.8)	17 (11.6)	5 (41.7)
	milk	2 (6.9)	8 (14.9)	50 (29.6)	15 (10.3)	6 (50.0)
	beer	2 (6.9)	12 (22.2)	47 (27.8)	14 (9.6)	5 (41.7)
	grain	3 (10.3)	4 (7.4)	34 (20.1)	14 (9.6)	6 (50.0)
	citrus	1 (3.4)	54 (100.0)	15 (8.9)	12 (8.2)	12 (100.0)
	not occurring in food	6 (20.7)	54 (100.0)	3 (1.8)	7 (4.8)	12 (100.0)
		Science fields *n* (%)				
The health effects in the human body caused by microplastics	inflammation	12 (41.4)	25 (46.3)	127 (75.1)	70 (47.9)	10 (83.3)
	thyroid diseases	16 (55.2)	26 (48.1)	100 (59.2)	60 (41.1)	8 (66.7)
	diseases of the digestive system	11 (37.9)	34 (63.0)	126 (74.6)	82 (56.2)	10 (83.3)
	disorders of the immune system	7 (24.1)	27 (50.0)	99 (58.6)	59 (40.4)	9 (75.0)
	cancers	11 (37.9)	36 (66.7)	134 (79.3)	97 (66.4)	10 (83.3)
	disorders of the reproductive system	5 (17.2)	17 (31.5)	79 (46.7)	43 (29.5)	7 (58.3)
	damage to vision and hearing	3 (10.3)	5 (9.3)	38 (22.5)	11 (7.5)	12 (100.0)
	osteoporosis	1 (3.4)	1 (1.9)	16 (9.5)	14 (9.6)	12 (100.0)
	dermatosis	29 (100.0)	54 (100.0)	25 (14.8)	14 (9.6)	12 (100.0)

* Answers to multiple choice question, answers do not add up to 100%.

**Table 4 nutrients-14-04857-t004:** Results of independence tests between knowledge of the topic of microplastics and data characterizing the study group.

Variables		*p*-Value	V-Cramer Coefficient
Definition of ‘microplastic’	Gender	0.002	0.19
	Field of study	<0.001	0.38
	Place of residence	0.001	0.23
	Professional status	0.1	-
Occurrence of microplastic in tap water	Gender	0.01	0.14
	Field of study	<0.001	0.30
	Place of residence	0.004	0.19
	Professional status	0.01	0.17
Occurrence of microplastic in bottled water	Gender	0.005	0.15
	Field of study	<0.001	0.31
	Place of residence	0.01	0.18
	Professional status	0.2	-
Occurrence of microplastic in food	Gender	0.4	-
	Field of study	0.1	-
	Place of residence	0.7	-
	Professional status	0.5	-
Vegetables most contaminated with microplastic	Gender	<0.001	0.23
	Field of study	<0.001	0.37
	Place of residence	0.1	-
	Professional status	0.4	-
Accumulation of microplastic in internal organs	Gender	0.5	-
	Field of study	0.2	-
	Place of residence	0.9	-
	Professional status	0.3	-

## Data Availability

The datasets generated and/or analyzed during the current study Microplastic in food and water. Current knowledge and awareness of consumers are not publicly available due to provisions of the data protection regulations.

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
