# Peer review of "Microplastic in Food and Water: Current Knowledge and Awareness of Consumers"

_nutrients, 2022, doi:10.3390/nu14224857_

Round 1

Reviewer 1 Report

The authors present an observational study (descriptive study) which aims to analyze to what extent consumers know about and are aware of the source of microplastics, the level of exposure, and potential health hazards connected to the contamination of food and water with microplastics. The test group, consisting of 410 people (Science fields: humanities, n =29; engineering and technical, n=54; medical and health, n=169; social sciences, n=146; natural sciences, n=12), is mostly able to correctly characterize what microplastics mean and knows its sources.

Comments:

1.

The test group is lot of subjects in medical and health and social sciences(85.4%) as well as 19-24 years(65.4%), a skewed distribution.

A representative population is important; data is generalisable if the research sample from which it is drawn is representative.

2.

Line 165-166,

78.3% of the question ot(to?)-methods had a very good agreement <0.75(?) or k> 0.75 denotes excellent reproducibility.??

3.

Table 3

The cells for “the occurrence of microplastics in food products” should be provided the number and percentage (n(%)).

4.

Table 4 presents a lot of repeat items, e.g., Gender, Field of study, Place of residence, Professional status. I would suggest to modify the Table.

Author Response

Dear Reviewer,

We sincerely thank you for your insightful analysis of our manuscript. Each of the comments was of great value to us, was thoroughly analyzed and found an appropriate response. Any changes have been highlighted in red. We believe that they will be sufficient to accept our manuscript for publication in Nutrients.

Once again, we thank you for your time.

The authors present an observational study (descriptive study) which aims to analyze to what extent consumers know about and are aware of the source of microplastics, the level of exposure, and potential health hazards connected to the contamination of food and water with microplastics. The test group, consisting of 410 people (Science fields: humanities, n =29; engineering and technical, n=54; medical and health, n=169; social sciences, n=146; natural sciences, n=12), is mostly able to correctly characterize what microplastics mean and knows its sources.

 Comments:

1.

The test group is lot of subjects in medical and health and social sciences(85.4%) as well as 19-24 years(65.4%), a skewed distribution.

A representative population is important; data is generalisable if the research sample from which it is drawn is representative.

RE: A cross-sectional study was presented by means of a diagnostic survey method. It was estimated that a sample of 410 students would be sufficient and representative of the Silesian region in Poland. It was assumed, according to the CSO (Central Statistical Office) Report, that the ratio of people studying in Silesia is 106411.Accordingly, on the formula: Nmin = NP ⋅ (α2 ⋅ f(1-f)) ÷ NP ⋅ e2 + α2 ⋅ f(1-f), where: Nmin - minimum sample size; NP - size of the population from which the sample is drawn; α - confidence level for the results; f - size of the fraction; e - assumed maximum error. The minimum sample size of students was calculated for the population of Silesia (Poland), which was 384 students (α = 0.95; f = 0.9; e = 0.05). Based on these calculations, the collected group of students was considered representative. Relationships between age, gender, and basic metric data between groups were examined, and the statistical test showed no relationship p>0.05.

2.

Line 165-166,

78.3% of the question ot(to?)-methods had a very good agreement <0.75(?) or k> 0.75 denotes excellent reproducibility.??

RE: In order to assess the reproducibility of the results obtained with the questionnaire used, the value of the χ parameter (Cohen's kappa) was calculated for each questionnaire question (results obtained in the baseline and retest). For 78.3% of the questions obtained a very good agreement Kappa>0.75.

3.

Table 3

The cells for “the occurrence of microplastics in food products” should be provided the number and percentage (n(%)).

RE: In the "Presence of microplastics in food products" cells, percentages were added. Both the question on the presence of microplastics in food products and the question on the health effects of microplastics in the body were multiple-choice questions.4.

Table 4 presents a lot of repeat items, e.g., Gender, Field of study, Place of residence, Professional status. I would suggest to modify the Table.

RE: In constructing Table 4, an effort was made to present the results of tests of independence between knowledge of the microplastic issue and the most important data characterizing the study group in as simple, clear and readable a manner as possible. Although many of the items in the table are repetitive, this procedure was necessary to enable proper interpretation of the data contained in the table.

Reviewer 2 Report

Line138-139:

Why did you only use students?

Figure 1:

Some legends are missing.

Line 252:

“mt” ?

Line 259:

Tale III? Is it Table 3?

Table 3:

For the question “The occurrence of microplastic in food products”, the two groups “Engineering and technical” and “Natural sciences” all have answered that it is not occurring in food, but at the same time they have also answered that it occurs in the different foods. That does not add up. Do you have an explanation?

The conclusion:

Please let the conclusion answer the hypothesizes.

Author Response

Dear Reviewer,

We sincerely thank you for your insightful analysis of our manuscript. Each of the comments was of great value to us, was thoroughly analyzed and found an appropriate response. Any changes have been highlighted in red. We believe that they will be sufficient to accept our manuscript for publication in Nutrients.

Once again, we thank you for your time.

Comments and Suggestions for Authors

Line138-139:

Why did you only use students?

RE: Taking this into account, in this study we decided to conduct an awareness survey of a population of students from the Silesian region (Poland) regarding the potential impact of microplastics on the risk of negative health effects. The choice of the study population was not random and was based on several important considerations. It is worth mentioning that the Silesian population living in a highly urbanized and industrialized area is particularly vulnerable to environmental health risks. The fundamental reason was that young consumers (and such are the vast majority of students) are expected to be more consumer and health conscious. The awareness in question at the same time is a determinant of their own health status, but in the near or long term, it will also shape the health status of their offspring [18, 19] . Sufficient consumer awareness makes it possible to understand the potential negative consequences of microplastic contamination of food, and to make informed, health-related consumer decisions. Young people, both educated in the field of medical and health sciences, but also educated in other areas of knowledge, should have a special perception of the health risks resulting from exposure to various xenobiotics, as evidenced by the large number of studies devoted to this issue. Studies are available on the perception of health risks resulting from exposure to air pollution or exposure resulting from heavy metals in food and water [20, 21, 22]. Since environmental pollution by microplastic particles is a relatively new issue, yet currently one of the most intensively researched, it was felt that investigating young people's knowledge of oral exposure to microplastics was warranted and would add to the, still scarce, body of knowledge in this area [23, 24, 25]. Finally, it is worth mentioning that the college-educated population is seen as particularly attractive for yet another reason. Well, due to the body of knowledge they present, the specificity of their duties resulting from receiving education, and familiarity with a research tool such as a questionnaire, students show a willingness to cooperate, see the validity of conducting scientific research and the possibility of using their results to improve the quality of life, in many dimensions. Thus, they are a particularly cooperative study group, which was seen as an additional potential in the selection of the study population.          

Figure 1:

Some legends are missing.

 RE: The legend next to Figure 2 is now complete. (Figure 1, currently Figure 2.)

Line 252:

“mt” ?

RE: "Mt" was an accidental editorial error. It has been removed.

Line 259:

Tale III? Is it Table 3?

RE:  Yes, the text refers to the data presented in Table 3. The error has been corrected.

Table 3:

For the question “The occurrence of microplastic in food products”, the two groups “Engineering and technical” and “Natural sciences” all have answered that it is not occurring in food, but at the same time they have also answered that it occurs in the different foods. That does not add up. Do you have an explanation?

RE: Indeed, when asked about the occurrence of microplastics in food products, respondents from two groups answered that microplastics are not found in food, while at the same time indicating food products contaminated with microplastic particles. Although these positions are in contrast to each other, their explanation is probably simple. Well, the question about food products contaminated with microplastic was a multiple-choice question. The answer "microplastic is not found in food" was the last possible choice ("j"). It is likely that some respondents, perhaps suggesting the correctness of previous answer suggestions, did not read all the possible answers and marked them unreflectively. In conclusion - the paradox highlighted in the review is probably an error due to the human factor, and at the same time evidence of the imperfection of the research tool, which is the survey questionnaire.

The conclusion:

Please let the conclusion answer the hypothesizes.

RE: The conclusions were modified accordingly and respond to the formulated research hypotheses.

Reviewer 3 Report

The article addresses the knowledge of a group of consumers about micoplastic. Both the introduction and a literature review should be aligned accordingly. The limitations mentioned by the authors are so crucial that no generalization of the results can be made. It is not even possible to find out exactly what the population of the investigation is.

Author Response

Dear Reviewer,

We sincerely thank you for your insightful analysis of our manuscript. Each of the comments was of great value to us, was thoroughly analyzed and found an appropriate response. Any changes have been highlighted in red. We believe that they will be sufficient to accept our manuscript for publication in Nutrients.

Once again, we thank you for your time.

Round 2

Reviewer 1 Report

No further comment

Reviewer 2 Report

I have no further comments.

Reviewer 3 Report

The changes made do not heal the shortcommings that I mentioned in the first review.